# Towards provably efficient quantum algorithms for large-scale machine-learning models

Junyu Liu [1,2,3,4,5,6], Minzhao Liu [7,8], Jin-Peng Liu [9,10,11], Ziyu Ye[2], Yunfei Wang[12], Yuri Alexeev[2,3,8], Jens Eisert [13] ✉ & Liang Jiang [1,3]

Large machine learning models are revolutionary technologies of artificial intelligence whose bottlenecks include huge computational expenses, power, and time used both in the pre-training and fine-tuning process. In this work, we show that fault-tolerant quantum computing could possibly provide provably efficient resolutions for generic (stochastic) gradient descent algorithms, scaling as $\mathcal{O}(T^2 \times \text{polylog}(n))$, where $n$ is the size of the models and $T$ is the number of iterations in the training, as long as the models are both sufficiently dissipative and sparse, with small learning rates. Based on earlier efficient quantum algorithms for dissipative differential equations, we find and prove that similar algorithms work for (stochastic) gradient descent, the primary algorithm for machine learning. In practice, we benchmark instances of large machine learning models from 7 million to 103 million parameters. We find that, in the context of sparse training, a quantum enhancement is possible at the early stage of learning after model pruning, motivating a sparse parameter download and re-upload scheme. Our work shows solidly that fault-tolerant quantum algorithms could potentially contribute to most state-of-the-art, large-scale machine-learning problems.

It is widely believed that large-scale machine learning might be one of the most revolutionary technologies benefiting society[1], including already important breakthroughs in digital arts[2], conversation like GPT–3[3,4], and mathematical problem solving[5]. However, training such models with considerable parameters is costly and has high carbon emissions. For instance, twelve million dollars and over five-hundred tons of $CO_2$ equivalent emissions have been produced to train GPT–3[6]. Thus, on the one hand, it is important to make large-scale machine-learning models (like large language models, LLM) sustainable and efficient.

On the other hand, machine learning might possibly be one of the flag applications of quantum technology. Running machine learning algorithms on quantum devices, implementing readings of so-called *quantum machine learning*, is widely seen as a potentially very fruitful application of quantum algorithms[7]. Specifically, many quantum approaches are proposed to enhance the capability of classical machine learning and hopefully find some useful applications, like[8,9]. Despite rapid development and significant progress, current quantum machine learning algorithms feature substantial limitations both in theory and practice. First, practical applications of quantum machine

[1]Pritzker School of Molecular Engineering, The University of Chicago, Chicago, IL 60637, USA. [2]Department of Computer Science, The University of Chicago, Chicago, IL 60637, USA. [3]Chicago Quantum Exchange, Chicago, IL 60637, USA. [4]Kadanoff Center for Theoretical Physics, The University of Chicago, Chicago, IL 60637, USA. [5]qBraid Co., Chicago, IL 60615, USA. [6]SeQure, Chicago, IL 60615, USA. [7]Department of Physics, The University of Chicago, Chicago, IL 60637, USA. [8]Computational Science Division, Argonne National Laboratory, Lemont, IL 60439, USA. [9]Simons Institute for the Theory of Computing, University of California, Berkeley, CA 94720, USA. [10]Department of Mathematics, University of California, Berkeley, CA 94720, USA. [11]Center for Theoretical Physics, Massachusetts Institute of Technology, Cambridge, MA 02139, USA. [12]Martin A. Fisher School of Physics, Brandeis University, Waltham, MA 02453, USA. [13]Dahlem Center for Complex Quantum Systems, Free University Berlin, Berlin 14195, Germany. ✉e-mail: jense@zedat.fu-berlin.de

learning algorithms for near-term devices are often lacking theoretical grounds that guarantee or at least plausibly suggest to outperform their classical counterparts. Second, for fault-tolerant settings of quantum machine learning problems[10–18], rigorous super-polynomial quantum speedups can actually be proven[19–21] for highly structured problems. That said, these prescriptions are arguably still far from real state-of-the-art applications of classical machine learning. Some of them are primarily using quantum states as training data instead of classical data, which can be—highly encouraging as these approaches are—argued to be not the currently most important classical machine learning application[20,22–25]. Efforts need to be made to extend our understanding of quantum machine learning, in the sense that we have to understand how they could have theoretical guarantees and how they could solve timely and natural problems, at least in principle, of classical machine learning. For instance, they should relate to scalable and sustainable natural problems in large-scale machine-learning.

In this work, we take significant steps in this direction by designing end-to-end quantum machine learning algorithms that are expected to be timely for the current machine learning community and that are to an extent equipped with guarantees. Based on a typical large-scale (classical) machine-learning process (see Fig. 1 for an illustration), we find that after a significant number of neural network training parameters have been pruned (sparse training)[26–29] and the classical training parameters compiled to a quantum computer, we suggest to find a quantum enhancement at the early state of training before the error grows exponentially. At its heart, the quantum algorithm part of the work includes suitable modifications of the quantum algorithm[30] for solving differential equations to running (stochastic) gradient descent algorithms—presumably the primary classical machine learning algorithm—into a quantum processor after linearization. The expectation of a possible quantum enhancement is rooted in an application of a variant of the so-called *Harrow-Hassidim-Lloyd* (HHL) algorithm[31], an efficient quantum algorithm for sparse matrix inversion that solves the problem within $\mathcal{O}(\log n)$ time for suitably conditioned $n \times n$ sparse matrices. We find that our algorithm can solve large-scale model-dimension-$n$ machine learning problems in $\mathcal{O}(\text{polylog}(n) \times T)$ or $\mathcal{O}(\text{polylog}(n) \times T^2)$ time, where $T$ is the number of iterations. The scaling in $n$ outperforms the scaling of any classical algorithms we know of. However, for a given machine learning problem with required performances, there is no guarantee that our hybrid quantum-classical algorithm will necessarily outperform all other conceivable classical algorithms for related, but different tasks (for instance, for algorithms that are not gradient-based). Thus, our result gives, to the best of our knowledge, rise to a potential substantial quantum speedup or enhancement of particular classical algorithms, instead of a quantum advantage over the entire problem class.

From a quantum algorithms perspective, stochastic gradient descent processes are solved here using quantum *ordinary differential equation* (ODE) solvers derived from the findings of ref. 30, based on linearizing non-linear equations using so-called quantum Carleman linearization. We find that the corresponding differential equation solvers can, in principle, also be used in the discrete setting and for stochastic gradient descent in machine learning. However, in the discrete setting, the theoretical details are significantly different from those applicable in the small learning rate limit. In this work, we systematically establish a novel discrete Carleman linearization in the supplementary material, including reformulations of the Carleman linearization theory, a tensor network diagrammatic notation for the discretization error, analytic derivations of higher-order corrections, and explicit examples for lower order expansions. Further details about the novelty of our algorithms beyond the findings of ref. 30 are summarized in the supplementary material.

It is important to stress that the above algorithm has a number of requirements that do admit a quantum enhancement. First, both the machine learning model and the weight vectors have to be sufficiently *sparse*, which will ensure a fast interface between classical and quantum processors (this requirement could be relaxed in the presence of *quantum random access memory* (QRAM)[32], a fast uploader towards quantum data, but we stress that this is *not* required and there are no hidden resources in our scheme). Second, the model has to be sufficiently *dissipative*. For dissipative systems, the linearization error is well controlled, ensuring that the HHL algorithm can obtain reliable results even with non-linear machine learning models. We find dissipation happens generically in the early training process of large-scale machine learning.

We corroborate the intuition developed here by a number of theorems, as well as extensive numerical experiments. The formal definition of dissipation, sparsity, and quantum speedups are rigorously proven in the supplementary material. Informal readings of the main theorems are presented in "Results", while solid numerical evidence up to 103 million training parameters are provided in "Numerical analysis". Finally, a conclusion providing also an outlook will be provided in "Discussion".

## Results
### Theorems
In this section, we will lay out the informally formulated main theorems that are established in this work. Details can be found in the supplementary material.

*Theorem 1* (Informal). For a sparse machine learning model with model size $n$, running $T$ iterations, with the algorithm being fully dissipative with small learning rates (whose formal definition is given in the supplementary material), there is a quantum algorithm that runs in

$$\mathcal{O}\left(T \times \text{poly}\left(\log n, \tfrac{1}{\epsilon}\right)\right) \qquad (1)$$

time with precision $\epsilon > 0$. The sparsity condition also ensures the efficiency of uploading and downloading quantum states towards classical processors.

*Theorem 2* (Informal). For a sparse machine learning model with model size $n$, running in $T$ iterations, and the algorithm being almost dissipative with small learning rates (whose formal definition is given in the supplementary material), then there is a

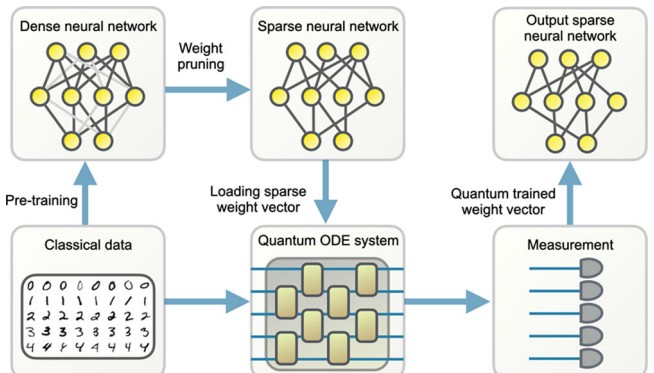

**Fig. 1 | A possible learning process in large-scale models, which might use sparse training, whose early stage in learning might admit possible quantum enhancement.** A dense neural network is pre-trained classically. The neural network weights are then pruned and only a small fraction is preserved. A quantum ordinary difference equation system that corresponds to the sparse training dynamics is created using the sparse network and the training data. To allow quantum enhancement, the system must be sparse and dissipative. Measurement on the solution state is performed to obtain the final trained parameters, used to construct a trained classical sparse neural network.

quantum algorithm that runs in

$$\mathcal{O}\left(T^2 \times \mathrm{poly}\left(\log n, \tfrac{1}{\epsilon}\right)\right) \qquad (2)$$

time with precision $\epsilon > 0$. The sparsity condition also ensures the efficiency of uploading and downloading quantum states towards classical processors.

In these expressions, $m := \log_2(n)$ takes the role of a system size of the quantum system. First, we describe the problem we are trying to solve. A machine learning model is defined partially by a function $\mathcal{L}_{\mathcal{A}}$, called the *loss function*, as a function of weight vector (variational angle) $\theta \in \mathbb{R}^n = (\theta_\mu)$, and the *input training set* $\mathcal{A}$. The weight vector has $n$ components if an $n$-dimensional model. The task is to minimize the function $\mathcal{L}_{\mathcal{A}}$ by adjusting $\theta$ making use of $T$ iterations.

The presumably most widely utilized algorithm in machine learning is called (stochastic) gradient descent. Starting from the initial weight vector $\theta(t=0)$, we implement the following ordinary differential equation from $t = 0$ to $t = T$ with small, positive learning rate $\eta$,

$$\theta_\mu(t+1) = \theta_\mu(t) - \eta \left.\frac{d\mathcal{L}_{\mathcal{A}}}{d\theta_\mu}\right|_{\theta(t)}. \qquad (3)$$

Variants of the gradient descent algorithms also include adding random noise $\xi_\mu(t)$ in each step, so-called stochastic gradient descent algorithms. One can show that in many cases, at the end of training, $\theta_\mu(t=T)$ can make the loss function $\mathcal{L}_{\mathcal{A}}(\theta(t=T))$ sufficiently small.

The quantum algorithm with the promised efficiency in Theorem 1 and Theorem 2 is described in the following.

- Our starting point of the algorithm is given by a initial weight vector, $\theta(0)$, the maximal number of iterations $T$, and the machine learning architecture $\mathcal{L}_{\mathcal{A}}$, with model size $n$.
- In a first step, we use so-called *quantum Carleman linearization* introduced in ref. 30, to linearize the model $\mathcal{L}_{\mathcal{A}}$ with the matrix $M$ (see the supplementary material for more details).
- Then, we need to upload the sparse weight vector $\theta(0)$ as a state vector in quantum devices, using tools of ref. 33 or alternatively more sophisticated and at the same time challenging architectures like *quantum random access memory* (QRAM)[32].
- Then, in a further step, we use a variant of the HHL solver that has been introduced in ref. 31 and the supplementary material, to solve the state vector at the end $t = T$. The pipeline is runnable under the condition of sparsity and dissipation, which is satisfied by our models. Sparsity includes the sparsity of model themselves, and the sparsity of weight vectors (ensured by the assumptions of sparse training), while dissipation is a natural property of the early steps of training, extensively discussed in "Numerical analysis".
- Finally, we exploit tomographic methods described in, for instance, refs. 24,34 and the supplementary material, to obtain the classical model parameters $\theta(T)$.

Finally, we wish to mention that this potential enhancement from quantum computing is concerning the size of the model, not necessarily the precision. For instance, we use tomographic method to download the sparse quantum state at the end of quantum ODE solver with the precision scaling as $1/\epsilon^2$. This scaling might be optimal in the quantum setting[34], but may not be ideal compared to purely classical algorithms (although the precise relationship between the error and the performances of classical machine learning models is generically not clear to date). We leave those interesting issues for future works.

## Linearizing classical neural networks

In this section, we provide a short and heuristic description of how to solve stochastic gradient descent using HHL algorithms. For a given

(stochastic) gradient descent process, the recursion relation is given by

$$
\begin{aligned}
\delta u = u(t+1) - u(t) &= \sum_{\ell=0}^{q} F_\ell u^{\otimes \ell}(t) \\
&= F_q u^{\otimes q}(t) + \ldots + F_2 u^{\otimes 2}(t) + F_1 u(t) + F_0 ,
\end{aligned}
\qquad (4)
$$

with the initial condition $u(0) = u_{\mathrm{in}}$. Here, $u(t) = (\theta_\mu)(t)$ is a set of weight vectors at the iteration $t$, $\delta o := o(t+1) - o(t)$ represents the discrete difference between two time steps $(t+1)$ and $(t)$ for a variable $o$ (see ref. 30 for the continuous version), and $u^{\otimes \ell}$ is the $\ell$-th order tensor product. Thus, Eq. (4) characterizes the dynamics of $q$-th order nonlinearity in classical neural networks. Now, we introduce a linear process designed to approximate the non-linear model (4), called *quantum Carleman linearization*, as

$$
\delta
\begin{bmatrix}
u \\
u^{\otimes 2} \\
u^{\otimes 3} \\
\vdots \\
u^{\otimes(N-1)} \\
u^{\otimes N} \\
\vdots
\end{bmatrix}
= A
\begin{bmatrix}
u \\
u^{\otimes 2} \\
u^{\otimes 3} \\
\vdots \\
u^{\otimes(N-1)} \\
u^{\otimes N} \\
\vdots
\end{bmatrix}
+
\begin{bmatrix}
F_0 \\
0 \\
0 \\
\vdots \\
0 \\
0 \\
\vdots
\end{bmatrix}.
\qquad (5)
$$

In this linear process, the vector space (whose vectors could be denoted by $\hat{y}$) is given by the weight vectors and all possible tensor products thereof, while $A$ is a large matrix with matrix elements given by the $F_\ell$, the so-called *quantum Carleman matrix* (QCM). In principle, this linear relation is infinitely dimensional, so we are replacing the original non-linear recursions to an infinite set of linear relations. If we wish to solve this infinite process by a digital system, we need to make truncation. In this work, we show that for dissipative systems (whose QCMs have enough negative eigenvalues, roughly corresponding to large enough positive eigenvalues for the Hessian of the loss functions in classical neural networks; the positive eigenvalues are called *dissipative modes* in the supplementary material, while negative eigenvalues are called *divergent modes*), the truncation error can be well-controlled.

For sparse, dissipative systems, Eq. (5) can be treated as a matrix inversion problem, thus solved by the HHL algorithm using quantum computers,

$$
\begin{bmatrix}
I & & & & \\
-(I+A) & I & & & \\
& \ddots & \ddots & & \\
& & -(I+A) & I & \\
& & & -(I+A) & I
\end{bmatrix}
\begin{bmatrix}
\hat{y}(0) \\
\hat{y}(1) \\
\vdots \\
\hat{y}(T-1) \\
\hat{y}(T)
\end{bmatrix}
=
\begin{bmatrix}
\hat{y}_{\mathrm{in}} \\
b \\
\vdots \\
b \\
b
\end{bmatrix}.
\qquad (6)
$$

Here, we are considering $T+1$ iterations in total from $t = 0$ to $t = T$, and the vector space has been further extended $T+1$ times. $I$ is the identity matrix, and $\hat{y}_{\mathrm{in}}$ is the initial weight vector corresponding to $u_{\mathrm{in}}$, written as a tensor product. This quantum ODE solver is our primary strategy towards solving stochastic gradient descent equations using quantum computers.

Although similar to its continuous version[30], the distinct differences between our algorithms and those of ref. 30 are extensively discussed in the supplementary material. More precisely, the discrete contributions will lead to higher order terms in the learning rate $\eta$. In fact, in the continuous case $A$ is linearly depending on various $F$, while in the discrete case, $A$ also has contributions scaling as $\eta^2 F^2 + \eta^3 F^3 \cdots$ (see the supplementary material for detailed examples). In the limit where $\eta \to 0$, the discrete contributions become identical to the

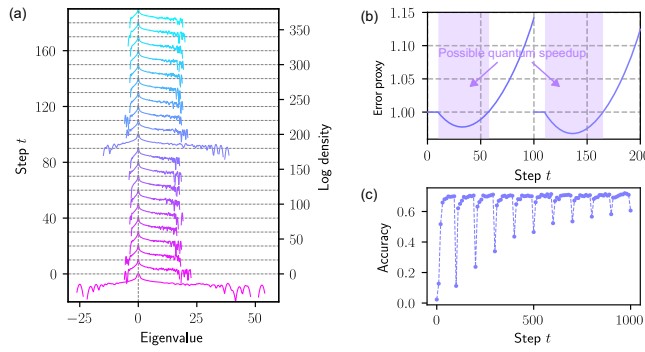

**Fig. 2 | Numerical results on ResNet as a function of step (Each step corresponds to a step of stochastic gradient descent based on the derivatives of the loss computed from 2048 randomly selected training samples). a** ResNet Hessian spectra during training. **b** Estimated error proxy during training. **c** Training accuracy evolution for ResNet.

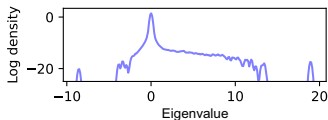

**Fig. 3 | Hessian of the pruned 103 million parameter model immediately after pruning without any additional training.**

continuous ones. It is also worth clarifying that those discrete contributions go beyond those considered in ref. 30. In fact, although one has to discretize the differential equations eventually in the ODE setup of ref. 30, the time derivative is computed before discretization for higher order Carleman linearization. For instance, $d(u^{\otimes 2})(t)$ is treated as $du \otimes u + u \otimes du$ both at the time $t$, while in the discrete case one has to consider contributions both from the $(t+1)$-st step and the $t$-th step. This contribution is the primary difference between the continuous and the discrete ODEs. Finally, in our problem setup, we always assume that an explicit form of the gradient descent equation is accessible, such that one can construct the Carleman linearization and make it available to quantum devices. This may not always be true for generic complicated classical neural networks whose complexity of analytic decomposition might grow with the size of the network. We leave a more thorough treatment to future research.

## Numerical analysis

In this part of our work, we focus on providing numerical evidence of a potential quantum enhancement for large-scale machine-learning models. Commercial large-scale LLMs like GPT-3 can have $\mathcal{O}(100)$ billion parameters and even more, which is challenging as a starting point due to its tremendous computational costs. Instead, here we provide examples of classification and computer vision machine learning models, which are relatively small compared to language models used in industry. Our computational resources allow us to achieve the scale up to $\mathcal{O}(100)$ million, which is both practically minded and reachable. We expect that LLMs and other models will feature a similar behavior to those examples since our algorithm works in general as a replacement for stochastic gradient descent.

Thus, in order to provide evidence of the functioning of our quantum algorithm in the context of practically minded machine learning, we perform numerical experiments on a state-of-the-art machine vision architectures, namely the so-called *ResNet*, to tentatively outline schemes with a potential quantum enhancement. First, we study a model with 7 million trainable parameters trained to distinguish images of 100 classes[35]. We first pre-train the neural network, use the largest 10% of learned parameters for initialization, and use the

quantum ODE system to obtain a sparse output model. We record the Hessian spectra during sparse training, allowing us to track the evolution of an error bound related quantity, given by

$$\frac{1}{N_c}\int_{-\infty}^{-0.4}\rho(a)|(1+a)^t|da+\frac{1}{N_c}\int_{0.4}^{\infty}\rho(a)|(1+a)^t|da, \quad (7)$$

where $\rho$ is the eigenvalue density, $a$ is the negative of Hessian eigenvalues, and $N_c$ is the renormalization constant implicitly defined by

$$\frac{1}{N_c}\int_{-\infty}^{-0.4}\rho(a)da+\frac{1}{N_c}\int_{0.4}^{\infty}\rho(a)da=1. \quad (8)$$

This error proxy discards small magnitude Hessian eigenvalues because they are close to 0, extremely abundant, and renders the error proxy stationary.

This numerical prescription is created according to criteria towards positivity of Hessian eigenvalues (dissipative modes). More dissipative systems have more positive Hessian eigenvalues, more negative $a$, and a better behaved error proxy. Specifically, the dissipative nature of the training dynamics initially leads to a reduction in this error proxy, which then gets overtaken by divergent modes and leads to an exponentially increasing error bound as shown in Fig. 2b. This motivates us to download the quantum trained model parameters sparsely and re-upload to the quantum computer to continue training every 100 steps. The effect of this procedure is that the exponentially increasing error restarts at 0 after re-uploading, with the side effect of Hessian broadening and accuracy reduction as shown in Fig. 2.

There is another strategy assuming the existence of QRAM. To combat the effect of Hessian broadening on the error proxy, we train the model classically for 10 steps after download before re-uploading of the new dense parameters, during which no training error is accrued. Although classical training has a cost linear in $n$, it is a small fraction of the entire training process. The accuracy dips immediately after download improves as training progresses, so our quantum training scheme is capable of producing useful sparse models. Finally, we examine the Hessian of a 103 million parameter ResNet. We start with a pre-trained model and prune 90% of the parameters. Due to the immense computational cost of computing Hessian for a large machine learning model (a relatively large-scale model for computational vision based on our computational resources), we only benchmark the Hessian spectra to provide evidences of dissipation and potential quantum enhancements. Figure 3 shows the initial Hessian, which clearly shows the dominance of dissipative modes over divergent modes similar to the 7 million parameter model. Since the Hessian improves with training for the 7 million parameter model, we believe this is evidence that the 103 million parameter model will have similarly manageable error growth.

## Discussion

In our work, we have provided quantum algorithm strategies that are presumably helpful for solving the (stochastic) gradient descent dynamics for large-scale classical machine learning models, like LLMs such as GPT-3. We identify certain precisely stated dissipative and sparse regimes of the model where quantum devices could meaningfully contribute, providing an end-to-end HHL-type quantum algorithm application that could outperform known classical algorithms. The observation that an efficient classical algorithm for efficiently solving all instances of non-linear dissipative differential equations would imply an efficient classical algorithm for any problem that can be solved efficiently by a quantum computer (is BQP hard)[30] can be seen as an argument that our algorithm is implausible to be de-quantized by classical proposals along the lines of ref. 36. Frankly, the core thesis of this work is that a main application of quantum computers may be in the *training of classical neural networks*.

Indeed, we claim that our algorithm might significantly increase the scalability and sustainability of classical large-scale machine-learning models and provide evidence for our claims numerically up to 103 million training parameters. Our work provides solid theoretical guarantees and intersections with state-of-the-art classical machine learning research. It sharply deviates from the mindset of variational quantum algorithms, and instead aims at augmenting classical machine learning by a key quantum step that constitutes a bottleneck for the classical training. In a way, it can be seen as adding flesh to the expectation that quantum formulations of neural networks may lead to new computational tools[37]. Specifically, our model requires the sparsity to be kept as a constant (or feature a polynomial scaling) in the size of the model to maintain a potential enhancement, which is consistent with the so-called *lottery ticket hypothesis*[38]. The setup is expected to be favorable in large-scale machine learning numerical experiments, although the sparsity ratio will generically decay.

Our work is expected to open up several potential directions in the field of quantum machine learning where one can reasonably hope for algorithmic improvements. In the supplementary material, we hint at a number of potentially particularly fruitful directions for future research. In short, they include the development of an alternative, time-dependent version during gradient descent trajectories, the identification of better formal criteria for dissipation, work on connections to diffusion models in classical machine learning and LLMs[39], theoretical improvements on the truncated HHL algorithms, and the identification of mechanisms of possible quantum speedups beyond notions of dissipation. We hope that this work can provide some stimulus for this type of research.

## Data availability
The full data for this work is available at ref. 40.

## Code availability
The full code for this work is available at ref. 40.

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

## Acknowledgements

We thank Senrui Chen, Vedran Dunjko, Aram Harrow, Robert Huang, Daliang Li, John Preskill, Jonah Sherman, Umesh Vazirani, Carl Vondrick, Han Zheng, Sisi Zhou and Peter Zoller for many valuable discussions. This research used the resources of the Argonne Leadership Computing Facility, which is a U.S. Department of Energy (DOE) Office of Science User Facility supported under Contract DE-AC02-06CH11357. J.L. is supported in part by International Business Machines (IBM) Quantum through the Chicago Quantum Exchange, and the Pritzker School of Molecular Engineering at the University of Chicago through AFOSR MURI (FA9550-21-1-0209). M.L. acknowledges support from DOE Q-NEXT. J.-P.L. acknowledges the support by the NSF (grant CCF-1813814, PHY-1818914), an NSF QISE-NET triplet award (DMR-1747426), an NSF QLCI program (OMA-2016245), a Simons Foundation award (No. 825053), and the Simons Quantum Postdoctoral Fellowship. Y.A. acknowledges support from DOE Q-NEXT and the DOE under contract DE-AC02-06CH11357 at Argonne National Laboratory. J.E. acknowledges funding of the ERC (DebuQC), the BMBF (Hybrid, MuniQC-Atoms), the BMWK (PlanQK, EniQma), the Munich Quantum Valley (K-8), the QuantERA (HQCC), the Quantum Flagship (Millenion, PasQuans2), the DFG (The Berlin Mathematics Research Center MATH+ (EXC-2046/1, project ID: 390685689), CRC 183), and the Einstein Research Foundation (Einstein Research Unit on Quantum Devices). L.J. acknowledges support from the ARO (W911NF-23-1-0077), ARO MURI (W911NF-21-1-0325), AFOSR MURI (FA9550-19-1-0399, FA9550-21-1-0209), AFRL (FA8649-21-P-0781), DoE Q-NEXT, NSF (OMA-1936118, ERC-1941583, OMA-2137642), NTT Research, and the Packard Foundation (2020-71479).

## Author contributions

J.L., M.L., J.-P.L., Z.Y., Y.A. and L.J. have initiated this work. All authors have contributed to pursuing the theoretical analysis and to proving the main claims. Y.W. has joined the team later for having made substantial contributions to the tensor network picture introduced. J.E. has in particular contributed to analysing sparsity and notions of tomographic recovery. J.L., M.L., J.-P.L., Z.Y. and Y.A. have performed the numerical analysis. All authors have contributed to discussions and to writing this manuscript.

## Funding

## Competing interests

The authors declare no competing interests.
