## [Peer Review File · Nature Communications]

Towards provably efficient quantum algorithms for large-scale machine-learning modelsREVIEWER COMMENTS

Reviewer #1 (Remarks to the Author):

The paper proposes a quantum algorithm to train machine learning models in the sparse regime where most of the weights are zero or close to zero. Under suitable assumptions, the runtime of the algorithm is polylogarithmic in the number of weights compared to the linear scaling of classical algorithms. Given the extremely large number of parameters of modern machine learning models, the algorithm could provide a major speedup and is extremely promising.

The proposed quantum algorithm considers the training of a machine learning model as a nonlinear differential equation for the time evolution of the vector of the weights and solves such equation with the quantum algorithm of Ref. [36] of the supplementary material. The novelty of the present paper with respect to Ref. [36] is the idea to apply the quantum algorithm for nonlinear differential equations proposed there to the training of machine learning models. Most of the technical results of the present paper are already present in Ref. [36]. Furthermore, besides the application to the training of machine learning models, the distinction between the new results and the results that are already present in Ref. [36] is not clear in the paper.

Therefore, I do not think that the paper in the present form is suitable for publication in Nature Communications.

Please find below my further comments about the paper:

- The proposed quantum algorithm to train machine learning models is efficient when the vectors of the weights is sparse both at initialization and after the training. However, also the classical training can become significantly faster if most of the weights are not trained and kept fixed to zero. It would be useful to discuss whether the proposed quantum algorithm is expected to keep its advantage also with respect to classical trainings that exploit sparsity.
- The proposed quantum algorithm requires the quantum Carleman matrix to be sparse. However, the paper does not prove that such assumption is satisfied by the cost function of deep neural networks, and therefore whether the proposed algorithm would actually be efficient is unclear.
- While the entire paper is based on the quantum algorithm of Ref. [36] of the

supplementary material, such reference does not appear in the main text, where the proposed algorithm is presented as completely new.

- The description of the quantum algorithm for nonlinear differential equations in section II of the supplementary material proceeds along the same lines as Ref. [36]. Presenting the algorithm of Ref. [36] can be useful to have a self-contained presentation. However, the distinction between the new results and the results that are already present in Ref. [36] should be made clear.
- The main text is not self-contained and some parts are not possible to understand without reading the supplementary material. For example, the main text mentions divergent modes without any description whatsoever, and eq. (4) appears completely out of the blue.
- Sec. I of the supplementary material contains several references to entities that are defined only in the following sections, thus forcing the reader to a continuous back and forth for a proper understanding.
- Since the matrix A is not Hermitian, in eq. (II.14) the inverse of the matrix S should appear rather than the conjugate transpose. Such mistake affects all the rest of Sec. II of the supplementary material.
- A derivation should be added for Eqs. (II.24) and (II.25).

Reviewer #2 (Remarks to the Author):

This paper presents technical and conceptual contributions in using quantum computers to assist in the training of classical machine learning models. The authors identify the resolution of SGD algorithms for sparse dissipative models as a potential area for superpolynomial quantum speedup, where the speedup appears in the dependence on model size. The authors present and collate technical tools which allow an “end-to-end” procedure for this problem.

In general both the main text and supplementary text are well written. The work is clear to specify the domain in which there is anticipated speedup.

Whilst the core routine has polylogarithmic dependence on approximation error, At present

it does not appear that the data uploading cost and tomographic cost are discussed concretely in the main text. The $1/\epsilon^2$ dependence on error in the tomographic cost is likely to be substantially worse than the error dependence of classical approaches, so I believe this should be clearly discussed in the main text. This is especially important if this work is to follow an “end-to-end” philosophy (for which I commend the authors).

If the ratio of pruned parameters is constant, then the sparsity of the output state scales linearly in n . Thus, the ratio of pruned parameters needs to increase sufficiently quickly with model size n to maintain polylogarithmic dependence of algorithm runtime with n . Firstly, I think this should definitely be indicated somewhere in the main text (as far as I can see, it does not appear to be there). Secondly, I have concerns that this requirement may affect performance for large model sizes and be a bottleneck. Could the authors comment on this?

The literature review seems very focused on quantum models. But here we have a classical machine learning task assisted via a quantum algorithm based on HHL. In this category, there are quite a few previous works such as [<https://arxiv.org/abs/1307.0471>, <https://arxiv.org/abs/1512.03929>]. I think this category of works are more relevant, and I think it would be useful to the reader to summarize how the present work fits in with this literature.

The authors state that there is hope that their algorithm is resistant to dequantization. The argument goes that one can consider the continuous limit with dissipativity going to zero, and it is known that simulating non-dissipative linear differential equations is BQP-hard. However, this work also has a sparsity assumption. In this context, one then thinks to another recent dequantization result for sparse matrices for a specific error scaling [<https://arxiv.org/abs/2111.09079>] and if it is directly applicable here, or at least gives concern that a similar result may hold in this context. Therefore, I am unsure if such a sweeping statement can be made without stating some caveats.

It is clear some nice work is done here. I think the manuscript would improve if some additional information could be indicated in the main text as mentioned above. I would also greatly appreciate if the authors could clarify some points, which would help me assess to a

finer degree the level of impact of this work. By clarifying such points also in the manuscript itself, I think the manuscript would be improved regardless of whether they sway favourably or unfavourably in terms of the impact of the paper.

Minor points:

In Theorem 1 and 2, should "Appendix" be "Supplemental material"?

At the bottom of Page 2, should "The quantum algorithm with the promised efficiency... described in the following" → "The quantum algorithm with the promised efficiency... is described in the following" ?

It seems Ref.[20] appears duplicated as Ref.[56]

Reply to Reviewer #1

Reviewer #1: *The paper proposes a quantum algorithm to train machine learning models in the sparse regime where most of the weights are zero or close to zero. Under suitable assumptions, the runtime of the algorithm is polylogarithmic in the number of weights compared to the linear scaling of classical algorithms. Given the extremely large number of parameters of modern machine learning models, the algorithm could provide a major speedup and is extremely promising. The proposed quantum algorithm considers the training of a machine learning model as a nonlinear differential equation for the time evolution of the vector of the weights and solves such equation with the quantum algorithm of Ref. [36] of the supplementary material. The novelty of the present paper with respect to Ref. [36] is the idea to apply the quantum algorithm for nonlinear differential equations proposed there to the training of machine learning models. Most of the technical results of the present paper are already present in Ref. [36]. Furthermore, besides the application to the training of machine learning models, the distinction between the new results and the results that are already present in Ref. [36] is not clear in the paper. Therefore, I do not think that the paper in the present form is suitable for publication in Nature Communications.*

Reply: We would like to start by sincerely thanking the referee for the report. We are delighted to read that “the algorithm could provide a major speedup and is extremely promising”. At the same time, it has been stressed clearly that we should emphasise more transparently the differences to the results of arXiv:2011.03185. We have taken this remark very seriously. In this reply letter, we explain in detail what we have done in order to fully accommodate the main comment. While it is true that we build upon the findings of arXiv:2011.03185, we go substantially beyond them. This has not become sufficiently clear in the previous version, we have corrected this in the manuscript and the supplemental material. We also explain it here in detail.

Firstly, obviously, our focus is markedly different: In our work, we target the dynamics of stochastic gradient descent in classical machine learning, whereas arXiv:2011.03185 is centering on solving ordinary differential equations. Making the leap from their work to ours is non-trivial, also in technical details, as we explain below. Secondly, the algorithm paralleling 2011.03185 is but one facet of our multifaceted hybrid quantum-classical approach. Beyond this, we offer a litany of novel contributions that distinctly surpass the current literature, including the following.

1. We extend the findings of arXiv:2011.03185 towards including general-order non-linearities, where the original literature is only applicable to second order.

2. We design and discuss novel pruning algorithms for classical neural networks suitable for the input-output problems for quantum devices.
3. We find that the key ingredients for quantum ODE solvers are sparsity and dissipation. We also provide quantitative definitions and theorems for dissipation conditions in classical neural networks (the Hessian matrices and their variants).
4. We devise novel tomography methods combining classical shadows and projective measurements, and relate them to the coupon collector's problem.
5. We perform large-scale numerical experiments up to 103 million training parameters, estimating the Hessian matrix spectra in those models, and verifying our statements. This is an aspect of our work that is a lot appreciated at conferences and usually leads to astounded reactions. We believe that we break the record of large-scale machine learning models in quantum literature. In fact, it could well be that this is the first scientific publication that makes use of industrial style classical data in machine learning.

Any of those points can already open up new research directions. In four aspects, we go substantially beyond the findings of arXiv:2011.03185.

But the referee is perfectly right in insisting that we should explain more clearly what we have technically done concerning the mentioned specific part of the algorithm. We deeply value the insights provided by the referee and editor, emphasizing the distinction between our work and arXiv:2011.03185. As a result of this endeavour, in our new version of the manuscript, we have enhanced our theories with new discrete variants of Carleman linearization, and also explain the differences better. Our supplementary material now encompasses the subsequent components:

1. We reformulate the Carleman linearization theory.
2. We show a tensor network diagrammatic notation for the discretization error.
3. We identify analytic derivations of higher-order corrections.
4. We present explicit examples for lower order expansions.

We have also highlighted and stressed those contributions on page 2 of the main text. We also explain in great detail the primary distinctions between our work and 2011.03185 extensively in the revised supplementary material.

Reviewer #1: *Please find below my further comments about the paper: The proposed quantum algorithm to train machine learning models is efficient when the vectors of the weights is sparse both at initialization and after the training. However, also the classical training can*

become significantly faster if most of the weights are not trained and kept fixed to zero. It would be useful to discuss whether the proposed quantum algorithm is expected to keep its advantage also with respect to classical trainings that exploit sparsity.

Reply: This is an excellent question, and we think we can provide a fair, even if not comprehensive answer to this question. Our claim is that we can outperform stochastic gradient descent methods for the pruned models in a super-polynomial fashion on a quantum computer. This is a safe claim. We say that “The observation that an efficient classical algorithm for efficiently solving all instances of non-linear dissipative differential equations would imply an efficient classical algorithm for any problem that can be solved efficiently by a quantum computer [as it is BQP complete] can be seen as a strong argument that our algorithm is implausible to be de-quantized by classical proposals [...]”. So here we can argue that while it is true that the pruning is making the classical step more efficient, we still get a separation to the quantum case. We think that this is a substantial step forward. There is also some good evidence that the known de-quantization methods based on l_2 sampling would not help here. What we cannot rule out, ultimately, that there are completely different classical algorithms that work in an entirely different fashion that in some way exploit the sparsity. At the end of the day, there is no separation proven between BQP and P, so it could—in principle—be that quantum computers are no more powerful than classical computers, implausible as this may sound. We have sharpened our precise claim in the manuscript which we think is a substantial improvement, and leave further studies to future work. We thank the referee for the good question.

Reviewer #1: *The proposed quantum algorithm requires the quantum Carleman matrix to be sparse. However, the paper does not prove that such assumption is satisfied by the cost function of deep neural networks, and therefore whether the proposed algorithm would actually be efficient is unclear.*

Reply: These are again good points. We do discuss several machine learning architectures in the supplementary material in Section III. We focus on two common classical settings, that of ResNet and that of MLP with bounded-polynomial activation functions. For ResNet, the derivatives of the neural networks are either 0 or 1 in the domain except the origin, so the Hessian spectra are constantly sparse, as we require them to be. For similar reasons, MLPs with bounded-polynomial activation functions give rise to similar properties. So the conditions of the quantum algorithms are satisfied. To corroborate this expectation, in our numerical examples, we are working with both models. Sparse neural networks are widely used in artificial intelligence and to the best of our knowledge they are pretty generic. We now emphasize those points in the main text on page 3.

Reviewer #1: *While the entire paper is based on the quantum algorithm of Ref. [36] of the supplementary material, such reference does not appear in the main text, where the proposed*

algorithm is presented as completely new.

Reply: In fact, the paper was indeed discussed and mentioned in the main text as Ref. [28] of the main text. But the referee is right in that we should stress this a lot and share the concern. So on page 2, we have added a paragraph containing an extensive discussion that elaborates on the differences of the quantum algorithm made use of here from that in Ref. [28].

Reviewer #1: *The description of the quantum algorithm for nonlinear differential equations in section II of the supplementary material proceeds along the same lines as Ref. [36]. Presenting the algorithm of Ref. [36] can be useful to have a self-contained presentation. However, the distinction between the new results and the results that are already present in Ref. [36] should be made clear.*

Reply: We thank the referee for pointing this out. For this reason, we have added a new section, Section II on page 3 and 4, in order to address this issue and to accommodate this wish. We have made an introductory description of the quantum Carleman linearization algorithm, and have emphasized the connection and differences compared to Ref. [36].

Reviewer #1: *The main text is not self-contained and some parts are not possible to understand without reading the supplementary material. For example, the main text mentions divergent modes without any description whatsoever, and eq. (4) appears completely out of the blue. Sec. I of the supplementary material contains several references to entities that are defined only in the following sections, thus forcing the reader to a continuous back and forth for a proper understanding.*

Reply: We have kept the main text on a high level, so that the broad readership understands the main lines of thinking and grasps the main results. To appeal to the expert readership, we offer this very comprehensive supplemental material. But the referee is right that the core ideas must be crystal clear also from reading the main text. For this reason, now the concepts related to divergent modes and Eq. (4) in the previous draft have been clarified an entirely the new Section II on pages 3 and 4 of the main text, and Section III around Eq. (8). Having added this material, the main text should now be self-contained.

Reviewer #1: *Since the matrix A is not Hermitian, in eq. (II.14) the inverse of the matrix S should appear rather than the conjugate transpose. Such mistake affects all the rest of Sec. II of the supplementary material.*

Reply: We thank the referee for pointing this out. We have modified them all throughout the text.

Reviewer #1: *A derivation should be added for Eqs. (II.24) and (II.25).*

Reply: We again thank the referee for this comment. A derivation has now been added.

We thank the referee for these positive comments and the constructive feedback. Again, we sincerely thank the referee a lot for insisting here. This feedback has helped us sharpening our presentation and spelling out substantially more clearly our technical contributions. This round of refereeing has been a constructive process. Having accommodated the core comment, as well as a number of smaller points, we hope that our work is now suitable for publication in its present form.

Reply to Reviewer #2

Reviewer #2: *This paper presents technical and conceptual contributions in using quantum computers to assist in the training of classical machine learning models. The authors identify the resolution of SGD algorithms for sparse dissipative models as a potential area for superpolynomial quantum speedup, where the speedup appears in the dependence on model size. The authors present and collate technical tools which allow an ‘end-to-end’ procedure for this problem. In general both the main text and supplementary text are well written. The work is clear to specify the domain in which there is anticipated speedup.*

Reply: We would like to thank the referee for the extraordinarily thoughtful report. We are delighted to read that both the main text and supplementary text are well written and that our work is clear to specify the domain in which there is anticipated speedup. We are glad about this overall positive stance towards our work. The referee adds a number of interesting and important comments, however, comments that we have taken very seriously. In the following, we spell out what we have done to accommodate the comments.

Reviewer #2: *Whilst the core routine has polylogarithmic dependence on approximation error, At present it does not appear that the data uploading cost and tomographic cost are discussed concretely in the main text. The $1/\epsilon^2$ dependence on error in the tomographic cost is likely to be substantially worse than the error dependence of classical approaches, so I believe this should be clearly discussed in the main text. This is especially important if this work is to follow an “end-to-end” philosophy (for which I commend the authors).*

Reply: We would like thank the referee for this comment. We agree with the referee’s statement, and we have discussed this issue now carefully around page 2, at the end of Section I. We believe that it should now be clear for the readers.

Reviewer #2: *If the ratio of pruned parameters is constant, then the sparsity of the output state scales linearly in n . Thus, the ratio of pruned parameters needs to increase sufficiently quickly with model size n to maintain poly-logarithmic dependence of algorithm runtime with n . Firstly, I think this should definitely be indicated somewhere in the main text (as far as I can see, it does not appear to be there). Secondly, I have concerns that this requirement may affect performance for large model sizes and be a bottleneck. Could the authors comment on this?*

Reply: These are good and reasonable points. We thank the referee for insightful comments. In fact, it is reasonable that the ratio of pruned parameters is decaying with the scaling of the model size. The reason is that the size of the problem solved by the classical machine learning model is fixed. The reasoning is rooted in the so-called lottery ticket hypothesis (J. Frankle and M. Carbin, “The lottery ticket hypothesis: Finding sparse, trainable neural networks,” in

International Conference on Learning Representations. 2019), which states roughly that for a given problem with the size m , and for a given classical machine learning model of size n , a small sparse subnetwork is expected to exist with the model size $\mathcal{O}(m)$ that has a comparable performance with the original model. Thus, it is acceptable to let the ratio decay with the size of the model, since the effective components of the network have the same size. So we believe that this observation does not hurt our algorithm generically. We discuss this in the conclusion section, on page 5 in the last two paragraphs.

Reviewer #2: *The literature review seems very focused on quantum models. But here we have a classical machine learning task assisted via a quantum algorithm based on HHL. In this category, there are quite a few previous works such as [<https://arxiv.org/abs/1307.0471>, <https://arxiv.org/abs/1512.03929>]. I think this category of works are more relevant, and I think it would be useful to the reader to summarize how the present work fits in with this literature.*

Reply: Once more, we thank the referee for the insightful comments. We have referenced and discussed this literature now in our introduction.

Reviewer #2: *The authors state that there is hope that their algorithm is resistant to dequantization. The argument goes that one can consider the continuous limit with dissipativity going to zero, and it is known that simulating non-dissipative linear differential equations is BQP-hard. However, this work also has a sparsity assumption. In this context, one then thinks to another recent dequantization result for sparse matrices for a specific error scaling [<https://arxiv.org/abs/2111.09079>] and if it is directly applicable here, or at least gives concern that a similar result may hold in this context. Therefore, I am unsure if such a sweeping statement can be made without stating some caveats.*

Reply: This is right, simulating non-dissipative linear differential equations is BQP-hard. The referee is also right that we are largely making use of a sparsity assumption, but by resorting to quantum random access memory, this can be relaxed. We are also aware of the beautiful work arXiv:2111.09079 and know it well: It proposes a dequantization algorithm for classes of quantum singular value transformations involving sparse matrices. They do so for arbitrarily small constant precision, which is less than we can achieve here. With inverse-polynomial precision, the same problem becomes BQP-complete. So that result is not directly applicable to our situation either, and we are confident that it will be challenging to dequantize the algorithm presented. But the referee is right and one should be careful. At the end of the day, it is not easy to rule out dequantization algorithms once and for all, not the least because there is no proven separation between BQP and P. For this reason, we have slightly lessened the claim.

Reviewer #2: *It is clear some nice work is done here. I think the manuscript would*

improve if some additional information could be indicated in the main text as mentioned above. I would also greatly appreciate if the authors could clarify some points, which would help me assess to a finer degree the level of impact of this work. By clarifying such points also in the manuscript itself, I think the manuscript would be improved regardless of whether they sway favourably or unfavourably in terms of the impact of the paper.

Reply: We thank the referee for the supportive feedback. Of course, we are happy to read that “some nice work is done here”. We have also taken the many additional comments very seriously. Indeed, it did help us sharpening our point and to stress the potential impact of our work much better. For this reason, we have discussed in great detail the differences with Ref. [34] and added new content concerning discrete versions of quantum Carleman linearization algorithms. On several occasions, we have also explained more clearly what has been achieved. The referee is very much right in insisting here. We sincerely hope that with our additional efforts, we have explained much better what we have done, and why there is strong evidence that we have an interesting and possibly practically relevant quantum algorithm here.

Reviewer #2: *In Theorem 1 and 2, should ‘Appendix’ be ‘Supplemental material’?*

Reply: This is right. We thank the referee for pointing this out and we have corrected it in the text.

Reviewer #2: *At the bottom of Page 2, should ‘The quantum algorithm with the promised efficiency described in the following’ → ‘The quantum algorithm with the promised efficiency is described in the following’?*

Reply: Again, the referee is right. Well spotted. We thank the referee for pointing this out and we have corrected it in the text.

Reviewer #2: *It seems Ref. [20] appears duplicated as Ref. [56]*

Reply: We could not detect this redundancy, actually.

Again, we thank the referee for this extremely careful and helpful report. This is not the first time we got such helpful and technically strong reports with the Nature Communications. To accommodate the requests, we have now explained much better what we have achieved in our work. We have also added technical material to present details in a clearer fashion. Overall, we hope that it has become manifest that we have an interesting quantum algorithm to offer, one with potential practical applications. Again, we sincerely thank the referee for the large amount of time invested in this. Having accommodated all comments, we hope that our work is now suitable for publication in its present form.

REVIEWER COMMENTS

Reviewer #1 (Remarks to the Author):

The Authors have successfully addressed most of my concerns. However, there are still some points that need to be further clarified before I can recommend the paper for publication:

- The Authors claim that a difference between the present paper and arXiv:2011.03185 is that the present paper considers a discrete version of Carleman linearization, while arXiv:2011.03185 considers Carleman linearization in continuous time. However, the algorithm of arXiv:2011.03185 is based on a discrete approximation of the continuous-time evolution equation, and it would be useful to have a comparison between the two discretizations
- It would be useful to add a section in the supplementary material to explain in more detail why ResNets and MLP with bounded-polynomial activation functions satisfy the sparsity assumptions required by the proposed algorithm

Reviewer #2 (Remarks to the Author):

I thank the authors for their thoughtful and extensive response to the referees' comments. Broadly, my questions have been clarified. My evaluation is that this is a wonderful thorough study of a potential setting for advantage for quantum algorithms, centered around a novel use case that the authors have identified. I am inclined to recommend this manuscript for acceptance.

I now understand that due to the Lottery Ticket Hypothesis one can hope for a sparse subnetwork of size scaling with the problem size. I appreciate the authors' detailed response about this, though I noticed the explanation they gave about the size of the sparse subnetwork does not seem to be anywhere in the text. It would be nice if this is also included.

It was a great idea to add the new section “Linearizing classical neural networks” — along with other improvements I believe this makes the key ideas easier to follow from the main text. One follow-up question I have is whether an analytic decomposition of the recursion relation is always easily accessible for popular models. It would be useful for the authors to briefly comment on this in the text. As I understand one would not usually go by such an approach in the classical setting?

For me there is still the open question of how the presented algorithm would perform against classical solvers which exploit sparsity. From the authors’ comments it seems this is not totally known at present. However, outside of this, the authors still present a quantum algorithm with self-contained exposition and clearly stated assumptions. I deem this to still be a great contribution, and a direction that should be highlighted and explored further.

Minor point:

- Should “ ∂o ” be “ ∂u ” after Eq.(4)?

Reply to Reviewer #1

The Authors have successfully addressed most of my concerns.

We would like to thank the reviewer for the kind feedback and for this assessment. Overall, this review process has been very productive so far and has helped us present our work in the best light.

However, there are still some points that need to be further clarified before I can recommend the paper for publication:

We have been more than happy to clarify these points.

- The Authors claim that a difference between the present paper and arXiv:2011.03185 is that the present paper considers a discrete version of Carleman linearization, while arXiv:2011.03185 considers Carleman linearization in continuous time. However, the algorithm of arXiv:2011.03185 is based on a discrete approximation of the continuous-time evolution equation, and it would be useful to have a comparison between the two discretizations

These are excellent and valid points. In fact, the discrete contributions will lead to higher order terms in the learning rate η . In fact, in the continuous case A is linearly dependent on various F , while in the discrete case, A also has contributions scaling as $\eta^2 F^2 + \eta^3 F^3 \dots$ (see the supplementary material for detailed examples). In the limit where $\eta \rightarrow 0$, the discrete contributions become identical with the continuous ones. It is also worth clarifying that those discrete contributions are beyond the results presented in arXiv:2011.03185, as we now also explain.

Actually, although one has to discretize the differential equations eventually in the ODE setup of arXiv:2011.03185, the time derivative is computed before discretization for higher order Carleman linearization. For instance, $d(u^{\otimes 2})(t)$ is treated as $du \otimes u + u \otimes du$ both at the time t , while in the discrete case one has to consider contributions both from the $(t+1)$ -th step and the t -th step. This contribution is the primary difference between the continuous and the discrete ODEs. We explain all those issues on page 4 of our manuscript. We have nothing to hide here, and the referee is right in insisting on seeing all details of these subtle and interesting points.

- It would be useful to add a section in the supplementary material to explain in more detail why ResNets and MLP with bounded-polynomial activation functions satisfy the sparsity assumptions required by the proposed algorithm.

We again thank the reviewer for the suggestion. In order to make it general, we have added an entirely new Section V E to the supplementary material, explaining that as long as the machine learning original model is well-pruned (like the example given in our paper), the overall sparsity of Quantum Carleman matrix A will not be affected. Thus, for pruned machine learning models the main theorems work. Some small details in the supplementary materials are modified

accordingly. Again, we thank the referee for these extraordinarily helpful comments. They have assisted us in presenting our results in the best possible fashion. Clarity is important, and it is also important to show that no hidden costs are swept under the carpet. Overall, one can say that this refereeing process has been enormously productive. This is how refereeing should work. Having accommodated the concerns, we hope our work is now suitable for publication.

Reply to Reviewer #2

I thank the authors for their thoughtful and extensive response to the referees' comments. Broadly, my questions have been clarified. My evaluation is that this is a wonderful thorough study of a potential setting for advantage for quantum algorithms, centered around a novel use case that the authors have identified. I am inclined to recommend this manuscript for acceptance.

We are delighted to read this positive assessment. This review process has been extraordinarily helpful and productive. It has led to a better manuscript that conveys the core message in a clearer fashion.

I now understand that due to the Lottery Ticket Hypothesis one can hope for a sparse subnetwork of size scaling with the problem size. I appreciate the authors' detailed response about this, though I noticed the explanation they gave about the size of the sparse subnetwork does not seem to be anywhere in the text. It would be nice if this is also included.

We thank the referee for this remark. This is an excellent point, we have added this in the Page 5 of our manuscript.

It was a great idea to add the new section "Linearizing classical neural networks" — along with other improvements I believe this makes the key ideas easier to follow from the main text. One follow-up question I have is whether an analytic decomposition of the recursion relation is always easily accessible for popular models. It would be useful for the authors to briefly comment on this in the text. As I understand one would not usually go by such an approach in the classical setting?

We thank the referee for pointing this out. For simple classical machine learning models (like the single-layer MLPs), the analytic decomposition might be easy to address, and we assume that the decomposition has been already done. For more complicated settings, the complexity of analytic decomposition might grow with the complexity of the network. Such a process is indeed discussed in work in progress by us, by combining either analog computing or further quantum linear algebra. We now discuss this on page 4 of the manuscript, and we have marked all changes again for clarity and transparency.

For me there is still the open question of how the presented algorithm would perform against classical solvers which exploit sparsity. From the authors' comments it seems this is not totally known at present. However, outside of this, the authors still present a quantum algorithm with self-contained exposition and clearly stated assumptions. I deem this to still be a great contribution, and a direction that should be highlighted and explored further.

We thank the referee for this assessment. Indeed, we completely agree here. In the new version, we have in even more detail explained how sparsity comes in technically in the quantum algorithm, and how it can be suitably exploited. Also, our algorithm seems robust against naive dequantizations resorting to standard methods. But at the end of the day, this does not mean that classical algorithms exploiting sparsity in a fresh and unexpected fashion are inconceivable. We have tried to portrait this in an honest and open way. In the end, we do believe that we have substantially contributed to the development of quantum algorithms for practically relevant problems here. And in the end, only the bigger community will be able to decide, a few years from now, which steps deserve further development and for which new, possibly quantum

inspired, algorithms can be used. We see this as a highly constructive and important research development.

Minor point:

- Should " ∂o " be " ∂u " after Eq. (4)?

We thank the referee for pointing this out. The change has been implemented. Again, we thank the reviewer for this extremely thoughtful and helpful report.

REVIEWERS' COMMENTS

Reviewer #1 (Remarks to the Author):

The Authors have addressed my concerns. As final suggestion, it would be useful to report the sparsity of the quantum Carleman matrices of the models studied in the numerical experiments.

Reviewer #2 (Remarks to the Author):

I thank the authors for their thoughtful responses and for addressing my remaining queries. I am happy to fully recommend this work for publication. Congratulations on the work.

Reply to Reviewer #1

We thank the referee for the comments. In our real example, the model sparsity is already 1%. However, obtaining exact numbers of sparsity in the quantum Carleman matrix is not computationally feasible in the example, since the number of parameters and the associated dimensions of the Carleman matrix is too large. That is what we explain in the Supplementary Material V.E, by showing that if the model itself is pruned well, then the quantum Carleman matrix will have good scalings in sparsity. Obtaining the exact number will require obtaining N , which we can only estimate and for which we can deliver complexity proofs rather than exact numbers. Considering that in the simulation, we are not using the original quantum Carleman matrix, we think for the point of illustrating behaviour of Hessians, the numerical calculation in the current paper is already sufficiently comprehensive (given that we use 7 million to 103 million training parameters, which to the best of our knowledge goes beyond any other similar simulation in the literature). That said, what the referee has pointed out is indeed valuable. In fact, some of the authors are working on a quantum architecture project, which we now mention in the draft.